



**Effects of ultraviolet radiation on photosynthetic performance and N2 fixation in**
***Trichodesmium erythraeum* IMS 101**
**Xiaoni Cai[1,2], David A. Hutchins[2], Feixue Fu[2] and Kunshan Gao[1]\***
[1]State Key Laboratory of Marine Environmental Science, Xiamen University, Xiamen,
Fujian, 361102, China
[2]Department of Biological Sciences, University of Southern California, 3616 Trousdale
Parkway, Los Angeles, California, 90089, USA
**Abstract**
Biological effects of ultraviolet radiation (UVR; 280–400 nm) on marine primary
producers are of general concern, as oceanic carbon fixers that contribute to the marine
biological $CO_2$ pump are being exposed to increasing UV irradiance due to global
change and ozone depletion. We investigated the effects of UV-B (280-320 nm) and
UV-A (320-400 nm) on the biogeochemically-critical filamentous marine $N_2$-fixing
cyanobacterium *Trichodesmium* (strain IMS101) using a solar simulator as well as
under natural solar radiation. Short exposure to UV-B, UV-A, or integrated total UVR
significantly reduced the effective quantum yield of photosystem II (PSII) and
photosynthetic carbon and $N_2$ fixation rates. Cells acclimated to low light were more
sensitive to UV exposure compared to high-light grown ones, which had more UV
absorbing compounds, most likely mycosporine-like amino acids (MAAs). After
acclimation under natural sunlight, the specific growth rate was lower (by up to 44%),
MAAs content was higher, and average trichome length was shorter (by up to 22%) in
the full spectrum of solar radiation with UVR, than under a photosynthetically active
radiation (PAR) alone treatment (400-700 nm). These results suggest that prior
shipboard experiments in UV-opaque containers may have substantially overestimated
in-situ nitrogen fixation rates by *Trichodesmium*, and that natural and anthropogenic
elevation of UV radiation intensity could significantly inhibit this vital source of new
nitrogen to the current and future oligotrophic oceans.





## Introduction

The stratospheric ozone depletion caused by anthropogenic inputs of chlorinated
fluorocarbons (CFCs) and other ozone-destroying substances have resulted in an
increase in ultraviolet radiation reaching the Earth's surface, especially UV-B radiation
(280-320 nm) (McKenzie et al., 2011). Additionally, global warming is inducing
shoaling of the upper mixed layer and enhancing stratification, thus exposing
phytoplankton cells which lived in the upper mixing layer to higher depth-integrated
irradiance (Häder and Gao, 2015). The increased levels of UV radiation especially UV-
B has generated concern about its negative effects on aquatic living organisms,
particularly phytoplankton, which require light for energy and biomass production.
Cyanobacteria are the largest and most widely distributed group of photosynthetic
prokaryotes on the Earth, and they contribute markedly to global $CO_2$ and $N_2$ fixation
(Sohm et al., 2011). Fossil evidence suggests that cyanobacteria first appeared during
the Precambrian era (2.8 to 3.5 $\times 10^9$ years ago) when the atmospheric ozone shield was
absent (Sinha and Häder, 2008). Cyanobacteria have thus often been presumed to have
evolved under more elevated UV radiation conditions than any other photosynthetic
organisms, possibly making them better equipped to handle UV radiation.
Nevertheless, a number of studies have shown that UV-B not only impairs the
DNA, pigmentation and protein structures of cyanobacteria, but also several key
metabolic activities, including growth, survival, buoyancy, nitrogen metabolism, $CO_2$
uptake, and ribulose 1,5-bisphosphate carboxylase activity (Rastogi et al., 2014). To
deal with UV stress cyanobacteria have evolved a number of defense strategies,
including migration to escape from UV radiation, efficient DNA repair mechanisms,
programmed cell death, the production of antioxidants, and the biosynthesis of UV-
absorbing compounds, such as MAAs and scytonemin (Rastogi et al., 2014; Häder et
al., 2015).
The non-heterocystous cyanobacterium *Trichodesmium* plays a critical role in the



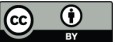

marine nitrogen cycle, as it is one of the major contributors to oceanic nitrogen fixation (Capone et al., 1997) and furthermore is an important primary producer in the tropical and sub-tropical oligotrophic oceans (Carpenter et al., 2004). This global importance of *Trichodesmium* has motivated numerous studies regarding the physiological responses of *Trichodesmium* to environmental factors, including visible light, phosphorus, iron, temperature, and $CO_2$ (Kranz et al., 2010; Shi et al., 2012; Fu et al., 2014; Spungin et al., 2014; Hutchins et al., 2015). However, to the best of our knowledge, nothing has been documented about how UV exposure may affect *Trichodesmium*.

*Trichodesmium* spp. have a cosmopolitan distribution throughout much of the oligotrophic tropical and subtropical oceans, where there is a high penetration of solar UV-A and UV-B radiation (Carpenter et al., 2004). It also frequently forms extensive surface blooms (Westberry and Siege, 2006), where it is presumably exposed to very high levels of UV radiation. Moreover, in the ocean, *Trichodesmium* populations may experience continuously changing irradiance intensities as a result of vertical mixing. Cells photoacclimated to reduced irradiance at lower depths might be subject to solar UVR damage when they are vertically delivered close to the sea surface due to mixing. Therefore, this unique cyanobacterium may have developed defensive mechanisms to overcome harmful effects of frequent exposures to intense UV radiation. Understanding how its $N_2$ fixation and photosynthesis respond to UV irradiance will thus further our knowledge of its ecological and biogeochemical roles in the ocean.

When estimating $N_2$ fixation using incubation experiments in the field, marine scientists have typically excluded UV radiation by using incubation bottles made of UV-opaque materials like polycarbonate (Capone et al., 1998; Olson et al., 2015). Thus, it seems possible that most shipboard measurements of *Trichodesmium* $N_2$ fixation rates could be overestimates of actual rates under natural UV exposure conditions in the surface ocean. In this study, *Trichodesmium* was exposed to spectrally realistic irradiances of UVR in laboratory experiments to examine the short-term effects of UVR on photosynthesis and $N_2$ fixation. In addition, *Trichodesmium* was grown under natural

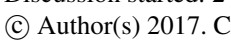


solar irradiance outdoors in order to assess UV impacts on longer timescales, and to test
for induction of protective mechanisms to ameliorate chronic UV exposure effects.

**Materials and methods**
**Study strategy** This study included two parts: (1) A short-term experiment under a
solar stimulator (refer to Fig.S1 for the spectrum) to examine the responses of
*Trichodesmium erythraeum* IMS 101 to a range of acute UV radiation exposures, and
(2) A long-term UV experiment under natural sunlight to examine acclimated growth
and physiology of *Trichodesmium* IMS 101. The first set of experiments was intended
to mimic intense but transitory UV exposures, as might occur sporadically during
vertical mixing, while the second set was intended to give insights into responses during
extended near-surface UV exposures, such as during a surface bloom event.
**Short-term UV experiment**      *Trichodesmium erythraeum* IMS101 strain was isolated
from the North Atlantic Ocean (Prufert-Bebout et al., 1993) was maintained in
laboratory stock cultures in exponential growth phase in autoclaved artificial seawater
enrich with nitrogen free YBCII medium (Chen et al., 1996). For the short-term UV
experiment, the cells were grown under low light (LL) 70 μmol photons $m^{-2}$ $s^{-1}$ and
hight light (HL) 400 μmol photons $m^{-2}$ $s^{-1}$ (12:12 light: dark) of PAR for at least 50
generations (about 180 days) prior to the UV experiments. These two light levels
represent growth sub-saturating and super-saturating levels for *Trichodesmium* (Cai et
al., 2015). Cultures were grown in triplicate using a dilute semi-continuous culture
method, with medium renewed every 4-5 days at 25°C. The cell concentration was
maintained at $< 5 \times 10^4$ cell $ml^{-1}$.

To determine the short-term responses of *Trichodesmium* IMS101 to UV radiation,

subcultures of *Trichodesmium* IMS101 were dispensed at a final cell density of 2-4 $\times$
$10^4$ cells $ml^{-1}$ into containers that allow transmission of all or part of the UV spectrum,
including 35 ml quartz tubes (for measurements of carbon fixation or measurements of



fluorescence parameters), 100 ml quartz tubes (for pigment measurements), or 13 ml
gas-tight borosilicate glass vials (for $N_2$ fixation measurements). Three triplicated
radiation treatments were implemented: (1) PAB (PAR+UV-A+UV-B) treatment,
using tubes covered with Ultraphan film 295 (Digefra, Munich, Germany), thus
receiving irradiances >295 nm; (2) PA (PAR+UV-A) treatment, using tubes covered
with Folex 320 film (Montagefolie, Folex, Dreieich, Germany), and receiving
irradiances >320 nm; and (3) P treatment: tubes covered with Ultraphan film 395 (UV
Opak, Digefra), with samples receiving irradiances above 395 nm, representing PAR
(400-700 nm). Since the transmission spectrum of the borosilicate glass was similar to
that of Ultraphan film 295, the borosilicate glass vials for $N_2$ fixation measurements of
PAB treatment were uncovered. Transmission spectra of these tubes (quartz and
borosilicate) and the various cut-off foils used in this study are shown in Fig. S1.
The experimental tubes were placed under a solar simulator (Sol 1200W; Dr. Hönle,
Martinsried, Germany) at a distance of 110 cm from the lamp, and maintained in a
circulating water bath for temperature control (25°C) (CTP-3000, Eyela, Japan).
Irradiance intensities were measured with a LI-COR $2\pi$ PAR sensor (PMA2100, Solar
light, USA) that has channels for PAR (400-700 nm), UV-A (320-400 nm) and UV-B
(280-320 nm). Measured values at the 110 cm distance were 87 $Wm^{-2}$ (PAR, ca. 400
µmol photons $m^{-2}$ $s^{-1}$), 28 $Wm^{-2}$ (UV-A) and 1 $Wm^{-2}$ (UV-B), respectively. For the
fluorescence measurements, samples were exposed under a solar stimulator for 60 min
and measurements of fluorescence parameters were performed during the exposure (see
below). Due to analytical sensitivity issues, for the carbon and $N_2$ incorporation
measurements, the exposure duration was 2 hrs, and for the measurements of UVAC
(UV-absorbing compounds) contents, the exposure time was 10 hrs.
**Long-term UV experiment**    To assess the long-term effects of solar ultraviolet
radiation on *Trichodesmium* IMS101, an outdoor experiment was carried during the
winter (Jan $1^{st}$ to Jan $26^{th}$, 2014) in subtropical Xiamen, China. 300-400 ml cell cultures
were grown in 500 ml quartz vessels exposed to 100% daytime natural solar irradiance





(surface ocean irradiance) (daytime PAR average of ~120W, highest PAR at noon
~300W). All of the quartz vessels were placed in a shallow water bath at 25°C using a
temperature control system (CTP-3000, Eyela, Japan). Two triplicated radiation
treatments were implemented: (1) treatment P: PAR alone (400-700 nm), tubes covered
with Ultraphan film 395 (UV Opak, Digefra); (2) treatment PAB: PAR+UV-A+UV-B
(295-700 nm), unwrapped quartz tubes. Incident solar radiation was continuously
monitored with a broadband Eldonet filter radiometer (Eldonet XP, Real Time
Computer, Möhrendorf, Germany) that was placed near the water bath. Daily doses of
solar PAR, UV-A and UV-B during the experiments are shown in Fig. S2. The
photoperiod during the outdoor incubation was 11:13 light:dark (light period from 7:00-
18:00 of local time). Cell were maintained in exponential growth phase (cell density <
$5 \times 10^4$), with dilutions (after sunset) every 4 days. All parameters were measured after
acclimation under respective P or PAB radiation for a week.

Specific growth rate ($\mu$, $d^{-1}$) of *Trichodesmium* IMS101 was determined based on

the change in cell concentrations over 4 days during the 8-11$^{th}$ and 12-15$^{th}$ day using
microscopic counts (Cai et al., 2015), the corresponding total dose from Day 8 to Day
11 and from Day 12 to Day 15 were 17.03 and 18.51 MJ, respectively. Chl *a* content
was measured at the 11$^{th}$, 15$^{th}$ and 19$^{th}$ day, and Chl *a*-specific absorption spectrum was
measured at the 18$^{th}$ day. Carbon and $N_2$ fixation rate were measured at 11:00-13:00 on
the 18$^{th}$ day; the diel solar irradiance record on that day is given in Fig. S3. In order to
separate the respective effects of UV-A and UV-B on carbon and $N_2$ fixation, a shift
experiment was carried out: subcultures from either P or PAB treatments were
transferred into another P (PAR), PA (PAR+UV-A), PAB (PAR+UV-A+UV-B)
treatment, which were marked as P′, PA′, PAB′ treatments, respectively. 35 ml quartz
tubes and 13 ml gas-tight borosilicate glass vials were used for carbon and $N_2$ fixation
measurements, respectively, as described below. Triplicate samples were used for each
radiation treatment for carbon and $N_2$ fixation, and the incubations were performed
under 100% solar irradiance for 2 hrs.



**Measurements and analyses**
**Effective photochemical quantum yield**     During the exposure under the solar
stimulator in the short-term experiment, small aliquots of cultures (2 ml) were
withdrawn at time interval of 3-10 min and immediately measured (without any dark
adaptation) using a Pulse-Amplitude-Modulated (PAM) fluorometer (Xe-PAM, Walz,
Germany). The quantum yield of PSII ($F_V'/F_M'$) was determined by measuring the
instant maximum fluorescence ($F_M'$) and the steady state fluorescence (Ft) under the
actinic light. The maximum fluorescence ($F_M'$) was determined using a saturating light
pulse (4000 μmol photons $m^{-2}$ $s^{-1}$ in 0.8 s) with the actinic light level set at 400 μmol
photons $m^{-2}$ $s^{-1}$, similar to the PAR level during the solar simulator exposure The
quantum yield was calculated as: $F_V'/F_M' = (F_M'-Ft)/F_M'$ (Genty et al., 1989).
**Chlorophyll-specific absorption spectra and UV-absorbing compounds (UVACs)**
Chl *a*-specific absorption spectra were measured on the 18[th] day, after consecutive
sunny days. Cellular absorption spectra were measured using the ''quantitative filter
technique'' (Kiefer and SooHoo, 1982; Mitchell 1990). The cells were filtered onto GF/F
glass fiber filters and scanned from 300 to 800 nm using a 1-nm slit in a
spectrophotometer equipped with an integrating sphere to collect all the transmitted or
forward-scattered light (i.e., light diffused by the filter and the quartz diffusing plate).
Filters soaked in culture medium were used as blanks. Chlorophyll-specific absorption
cross-sections (a*) were calculated according to Cleveland and Weidemann (1993) and
Anning et al., (2000). Content of Chl *a* and UV-absorbing compounds (UVACs) were
measured by filtering the samples onto GF/filters and subsequently extracted in 4 mL
of 100% methanol overnight in darkness at 4 °C. The absorption of the supernatant was
measured by a scanning spectrophotometer (Beckman Coulter Inc., Fullerton, CA,
USA). The concentration of Chl *a* was calculated according to Ritchie (2006). The main
absorption values for UV-absorbing compounds ranged between wavelengths of 310
and 360 nm, and the peak absorption value at 332 nm was used to estimate total
absorptivity of UVACs according to Dunlap et al., (1995). The absorptivity of UVACs



was finally normalized to the Chl *a* content (μg (μg Chl *a*)$^{-1}$).
*Trichodesmium* IMS101 UVACs content was compared to that of three other
marine phytoplankton species, including *Chlorella*.sp, *Phaeodactylum tricornutum*,
and *Synechococcus* WH7803, representing a green alga, a diatom and a unicellular
cyanobacterium, respectively. All cultures were maintained under the same conditions
(25$^{o}$C, 150 μmol photons m$^{-2}$ s$^{-1}$) for several days prior to pigment extraction. The
absorption spectra were measured by filtering the samples on GF/filters that were
subsequently extracted in 4 mL of 100% methanol overnight at 4 $^{o}$C. The absorption
spectra of the supernatant were scanned from 250 to 800 nm in a spectrophotometer
(Beckman Coulter Inc., Fullerton, CA, USA). The Optical Density (OD) values were
then normalized to OD (662 nm), Chl *a* peak.
**Carbon fixation rates**    Carbon fixation rate of both short- and long-term experiments
were measured using the $^{14}$C method. A total of 20 ml samples were placed in 35 ml
quartz tubes and inoculated with 5μCi (0.185 MBq) of labeled sodium bicarbonate (ICN
Radiochemicals), and were then maintained under the corresponding radiation
treatments for 2 hrs. After incubation, the cells were filtered onto Whatman GF/F filters
(Φ 25 mm) and stored at -20$^{o}$C until analysis. To determine the radioactivity, the filters
were thawed and then exposed to HCl fumes overnight and dried at 60$^{o}$C for 4 hrs
before being placed in scintillation cocktail (Hisafe 3, Perkin-Elmer, Shelton, CT, USA),
and measured with a scintillation counter (Tri-Carb 2800TR, Perkin-Elmer, Shelton,
CT, USA) as previously described (Cai et al., 2015).
**N$_2$ fixation rates**    Rates of N$_2$ fixation for both short- and long-term experiments were
measured in parallel with the carbon fixation measurements using the acetylene
reduction assay (ARA) (Capone et al., 1993). Samples of 5 ml subcultures were placed
in 13 ml gas-tight borosilicate vials (described above), and 1ml acetylene was injected
into the headspace before incubating for 2 hrs under the corresponding radiation
treatment conditions. A 500 μl headspace sample was then analyzed in a gas





chromatograph equipped with a flame-ionization detector and quantified relative to an ethylene standard. The ethylene produced was calculated using the Bunsen gas solubility coefficients according to Breitbarth et al., (2004) and an ethylene production to $N_2$ fixation conversion factor of 4 was used to derive $N_2$ fixation rates, which were then normalized to cell number.

**Data analysis** The inhibition of $\Phi$PSII, carbon fixation and $N_2$ fixation due to UVR, UV-A, or UV-B was calculated as:

UVR-inducted inhibition = $(I_P - I_{PAB})/I_P \times 100\%$

UV-A-inducted inhibition = $(I_P - I_{PA})/I_P \times 100\%$

UV-B-inducted inhibition = $UVR_{inh} - UVA_{inh}$

where $I_P$, $I_{PA}$, $I_{PAB}$ indicate the values of carbon fixation or $N_2$ fixation in the P, PA and PAB treatments, respectively. Repair (r) and damage (k) rates during the 60 min exposure period in the presence of UV were calculated using the Kok model (Heraud and Beardall, 2000):

$P/P_{initial} = r/(r+k) + k/(r+k) \times \exp(-(r+k) \times t)$,

where $P_{initial}$ and $P$ were the yield values at the beginning and at exposure time t. Three replicates for culture conditions or each radiation conditions were used in all experiments, and the data are plotted as mean and standard deviation values. Two way ANOVA tests were used to determine the interaction between culture conditions and UVR at a significance level of p=0.05.

**Results**

**Short-term UV experiment** The effects of acute UVR exposure on cells grown under LL and HL conditions are shown in Fig.1. For the cells grown under LL condition, the $F_V'/F_M'$ declined sharply within 10 min after first exposure in all radiation treatments,





and then leveled off. $F_V'/F_M'$ decreased less in the samples receiving PAR alone (to 43%
of the initial value) than those additionally receiving UV-A (to 30% of the initial value)
or UV-A+UV-B (to 24% of the initial value) (Fig.1A). The $F_V'/F_M'$ value of PA and
PAB treatments were significantly lower compared to the PAR treatment (p=0.03 and
p<0.01, respectively). $F_V'/F_M'$ of HL grown cells declined less and more slowly
compared to the LL grown cells. The $F_V'/F_M'$ of HL cells under PAR alone remained
more or less constant during the exposure, since the PAR level was similar to the growth
level of HL (400 μmol photons $m^{-2}$ $s^{-1}$). In contrast, the $F_V'/F_M'$ decreased to 75% and
65% of its initial value for the PA and PAB treatment, respectively, and were
significantly lower than PAR treatment (p<0.01) (Fig.1B).

The damage and repair rates of the PSII reaction center estimated from the

exponential decay in the effective quantum yield showed higher damage and lower
repair rates in the LL-grown cells than in the HL-grown ones (Fig.1C,D). The PSII
damage rates (k, $min^{-1}$) of LL grown cells were 0.14, 0.16 and 0.15 $min^{-1}$ in the P, PA
and PAB treatments, respectively, about 2 times faster than in the cells grown under HL
conditions (Fig.1C). The PSII repair rates (r, $min^{-1}$) of LL grown cells were 0.1, 0.06
and 0.05 $min^{-1}$ in the P, PA and PAB treatments, which were 83% (p<0.01), 33% (p<0.01)
and 54% (p<0.01) lower than in HL grown cells, respectively (Fig.1D). The damage
rate was not significantly different among P, PA and PAB treatment within either of
the LL- and HL-grown treatments (p>0.05), but the repair rate was much higher in the
P treatment without UV than in PA or PAB treatments in the HL-grown cells (p<0.01).

The photosynthetic carbon fixation and $N_2$ fixation rates during the UV exposure

are shown in Fig. 2. The HL-grown cells had 17% higher photosynthetic carbon fixation
rates than the LL-grown ones under the PA treatment (p<0.01), however, the LL and
HL-grown cells didn't show significant differences in carbon fixation rates under the P
and PAB treatments (p=0.29, and p=0.06). In the presence of UV radiation, carbon
fixation was significantly inhibited in both LL and HL-grown cells (Fig.2A). Carbon
fixation inhibition induced by UV-A was about 35-45%, much larger than that induced



by UV-B, which caused only about a 10% inhibition of carbon fixation (p<0.01). The
UV-A exposed carbon fixation rate was significantly higher in the LL- grown cells than
in HL grown cells (p<0.01), while UV-B did not cause a significant difference in
inhibition between the HC- and LC-grown cells (p=0.88) (Fig. 2B). $N_2$ fixation rates
were about twofold higher in HL-grown cells in all radiation treatments (Fig.2C,
p<0.01), but the UV-induced $N_2$ fixation inhibition showed no significant differences
between the LL and HL grown cells regardless of UV-A or UV-B exposures (Fig. 2D,
p=0.80, 0.62, 0.39 for UVA-, UVB-, and UVR-induced inhibition, respectively).
Compared to other phytoplankton under the same growth conditions,
*Trichodesmium* IMS101 had much higher absorbance in the UV region (300-400 nm)
(Fig. 3A). In this study, the absorbance at 332 nm of HL-grown cells was about twofold
higher compared to LL-grown ones (Fig. 3B). However, the cellular Chl *a* content (data
not shown) and UVACs contents of both LL and HL grown cells did not change after
exposure to UV for 10 hrs (Fig. 3C).
**Long-term UV experiment** After being acclimated under full natural solar radiation
for 7 days, the specific growth rates of cells grown under the PAB treatment were
0.15±0.01 and 0.14±0.06 during the 8-11$^{th}$ day and 12-15$^{th}$ day periods, respectively.
These growth rates were significantly lower by 44% and 39% compared to cells grown
under the P treatment, respectively (Fig.4A, p=0.014 and p=0.03). The mean trichome
lengths of PAR treatment cells on the 11$^{th}$ and 15$^{th}$ day were 758±56 and 726±19 μm,
while addition of UVR significantly reduced the trichome length by 22% and 11%
(p=0.02 and p=0.02).
Analysis of the Chl *a* specific absorption spectra, a*(λ), demonstrated that UVR
had a major effect on the absorbance of UV regions and phycobilisomes (Fig. 5). The
optical absorption spectra revealed a series of peaks in the UV and visible wavelengths
corresponding to the absorption peaks of UVACs at 332 nm, Chl *a* at 437 and 664 nm,
phycourobilin (PUB) at 495 nm, phycoerythrobilin (PEB) at 545 nm,





phycoerythrocyanin (PEC) at 569 nm, and phycocyanin (PC) at 627 nm. In the UV
region, the a*($\lambda$) value was higher in the PAB treatment cultures than in the P treatment
cultures (Fig. 5). The UVR treatments did not show clear effects on Chl *a* content
compared to acclimation to PAR alone measured on different days (Fig. S3). However,
the ratio of UVACs to Chl *a* was increased by 41% in the PAB compared to the P
treatment ($p < 0.01$).

The cells grown in the long-term P and PAB treatments showed different responses

for carbon and $N_2$ fixation after being transferred to short-term P′, PA′, and PAB′
radiation treatments at noon on the 18th day (Fig. 6). P and PAB acclimated cells did
not show significant differences in carbon fixation among all short-term P′, PA′, PAB′
treatments (Fig. 6A, $p = 0.17$, $p = 0.22$, $p = 0.51$, respectively), nor in the UV-induced
inhibition of carbon fixation (Fig. 6B, $p > 0.05$). However, long-term UV-A exposure
inhibited short-term carbon fixation by about 58% in both the P and the PAB treatments,
significantly higher than that induced by UV-B radiation (Fig. 6B, $p < 0.01$).

$N_2$ fixation rates of P acclimated cells were significantly higher than PAB

acclimated cells in all P′, PA′, and PAB′ treatments (Fig. 6C, $p < 0.01$). The $N_2$ fixation
inhibition induced by UV-A of PAB acclimated cells was 49%, significantly higher by
47% than that of P acclimated cells ($p = 0.03$), while there was no significant difference
in UVB-induced $N_2$ fixation inhibition between P and PAB acclimated cells (Fig. 6D,
$p = 0.62$). The carbon fixation rates measured under PAR (PAR treated cells to P') and
PAB (PAB treated cells to PAB') conditions were 89.2 and 47.1 fmol C cell$^{-1}$ h$^{-1}$,
respectively, while $N_2$ fixation rates measured under those conditions were 1.9 and 0.5
fmol $N_2$ cell$^{-1}$ h$^{-1}$. UVR exposure lowered estimates of carbon and $N_2$ fixation rates by
47% and 65%, respectively.

**Discussion**

Our study shows that growth, photochemistry, photosynthesis and $N_2$ fixation in





*Trichodesmium*.sp are all significantly inhibited by UVR, including both UV-A and UV-
B. These effects occur in both short-term, acute exposures, as well as after extended
exposures during acclimated growth. These results are ecologically relevant, since this
cyanobacterium is routinely exposed to elevated solar irradiances in its tropical habitat
either transiently, during vertical mixing, or over longer periods during surface blooms.
*Trichodesmium* provides a biogeochemically-critical source of new N to open ocean
food webs, so significant UV inhibition of its growth and $N_2$ fixation rates could have
major consequences for ocean biology and carbon cycling.
Short exposure to UVR causes a significant decline in the quantum yield of
photosystem II (PSII) fluorescence of *Trichodesmium*, that is consistent with damage
to critical PSII proteins such as D1 in a brackish water cyanobacterium *Arthrospira*
*(Spirulina) platensis* (Wu et al., 2011). UV-induced degradation of D1 proteins results
in inactivation of PSII, leading to reduction in photosynthetic activity (Campbell et al.,
1998). In addition, studies of various microbial mats have shown that Rubisco activity
and supply of ATP and NADPH are inhibited under UV exposure, which might also
lead to the reduction in photosynthetic carbon fixation (Cockell and Rothschild, 1999;
Sinha et al., 1996, 1997).
Exposure to UVR had an impact on nitrogenase activity in *Trichodesmium*, since
both the short- and the long-term UV exposure led to significant reduction of $N_2$ fixation
of up to 30% (short-term) or ~60% (long-term) (Fig. 2D and 6D). Studies on the
freshwater cyanobacterium *Anabaena*. sp. showed a 57% decline in $N_2$ fixation rate
after 30min exposure to UVR of 3.65W (Lesser, 2007). Some rice-field cyanobacteria
completely lost $N_2$ fixation activity after 25-40 min exposure to UV-B of 2.5 W (Kumar
et al., 2003). In our results, long-term exposure to UV led to higher inhibition of $N_2$
fixation, implying that accumulated damage to the key $N_2$-fixing enzyme, nitrogenase,
could have occurred during the growth period under solar radiation in the presence of
UVR.





Compared to $N_2$ fixation, UVR induced an even higher degree of inhibition of
carbon fixation. The carbon fixation rate decreased by 50% in the presence of UVR.
UV-A induced higher inhibition than UV-B, indicating that although UV-B photons
(295-320 nm) are in general more energetic and damaging than UV-A (320-400 nm),
the greater fluxes of UV-A caused more inhibition of carbon fixation, which was
consistent with other studies of spectral dependence of UV effects (Cullen and Neale
1994; Neale 2000). This finding is ecologically significant, since UV-A penetrates
much deeper into clear open ocean and coastal seawater than does UV-B.
Compared to low light-grown cells, the high light-grown ones were more resistant
to UVR, which was reflected in the lower PSII damage rate and faster recovery rate in
the presence of UVR, as well as the significantly lower levels of carbon fixation
inhibition caused by UV-A and/or UV-B. Such a reduced sensitivity to UVR coincided
well with a significant increase in UV-absorbing compounds in the HL-grown cells
compared to the LL-grown ones. Similar dependence of photosynthetic sensitivity to
UV inhibition on growth light levels has been reported in other species of
phytoplankton (Litchman and Neale, 2005; Sobrino and Neale, 2007). The sensitivity
of PSII quantum yield to UV exposure in *Synechococcus* WH7803 was also less in
high-light-grown versus low-light-grown cells (Garczarek et al., 2008). In addition, it
has been observed that phytoplankton from turbid waters or acclimated to low-light
conditions are more sensitive to UVR than those from clear waters (Villafane et al.,
2004; Litchman and Neale, 2005; Helbing et al., 2015). These observations suggest that
*Trichodesmium* sp. may acclimate to growth in the upper mixed layer by producing UV-
absorbing compounds, making them more tolerant of UVR than cells living at deeper
depths.
Although UV radiation can clearly cause damage to PSII and inhibit physiological
processes in *Trichodesmium* sp., this cyanobacterium has evolved protective
biochemical mechanisms to deal with UV radiation in their natural high-UV habitat.
One important class of UV-absorbing substances are mycosporine-like amino acids

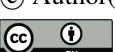



(MAAs) and scytonemin. These compounds strongly absorb in the UV-A and/or UV-B
region of the spectrum, and dissipate its energy as heat without forming reactive oxygen
species, protecting the cells from UV and from photooxidative stress (Banaszak 2003).
The ''mycosporine-like amino acids'' (MAAs), which have strong UV-absorption
maxima between 310 and 362 nm (Sinha and Häder, 2008) as identified by HPLC in
other studies, consist of a group of small, water-soluble compounds, including asterina-
332 ($\lambda$max=332) and shinorine ($\lambda$max=334), which are the most abundant, as well as
mycosporine-glycine ($\lambda$max=310), porphyra-334 ($\lambda$max=334), and palythene
($\lambda$max=360) (Shick and Dunlap 2002; Subramaniam et al., 1999). As was found
previously in *Trichodesmium* spp., high absorbance in the UV region is mainly due to
the presence of ''mycosporinelike amino acids'' (MAAs), with absorbance maxima
between 310~362 nm (Sinha and Häder, 2008).
Our investigation strongly suggests that *Trichodesmium* is able to synthesize
MAAs ($\lambda$max ~330 nm and 360 nm) in response to elevated PAR and UV radiation.
Synthesis of MAAs has been reported to be stimulated by high PAR and UV radiation
in other phytoplankton (Karsten et al., 1998; Vernet and Whitehead, 1996; Sinha et al.,
2001). Our high light-grown cells were more tolerant of UVR, likely at least partly due
to their ability to synthesize double the amount of MAAs in comparison to low light-
grown ones (Fig.3B). It has been showed that accumulation of MAAs may represent a
natural defensive system against exposure to biologically harmful UV radiation
(Karsten et al., 1998) and cells with high concentrations of MAAs are more resistant to
UVR than cells with small amounts of these compounds (Garcia-Pichel and Castenholz,
1993). In fact, MAAs concentrations varying between 0.9 and 8.4 ug mg (dry weight)$^{-}$
$^{1}$ have been measured in cyanobacterial isolates (Garcia-Pichel and Castenholz, 1993),
and ratios of MAAs to Chl *a* in the range from 0.04 to 0.19 have been reported in
cyanobacterial mats (Quesada et al., 1999). In our study, we found that *Trichodesmium*
contained a much higher concentration of MAAs (the highest value in HL-grown cells
is 5 pg cell$^{-1}$) and that the ratio of these compounds to Chl *a* was 5, consisted with



previous reports in regard to *Trichodesmium* (Subramaniam et al., 1999), which is much
higher than in other phytoplankton. This adaptation could be a major reason for the
ability of *Trichodemium* to grow and form extensive surface blooms under strong
irradiation in the oligotrophic oceans.

In our study, no significant changes in the amount of MAAs were observed after

10 h of exposure to UVR under the solar simulator. In contrast, a significant increase
of 23% in the concentration of MAAs was observed in full solar spectrum treated cells
compared to PAR-treated ones grown outdoors after consecutive sunny days (on the
18[th]). It seems that the synthesis of MAAs takes a relatively long time. Other studies
have shown the time required for induction of MAAs in other cyanobacteria is
dependent on UV doses and species, and shows a circadian rhythm (Sinha et al., 2001;
Sinha et al., 2003).

Not only did long-term exposure to high solar UV radiation significantly reduce

*Trichodesmium*'s growth rate (by 37~44%), but it also significantly shortened its
average trichome length (less cell per filament) (Fig. 4). The decreased growth rates
correlated with decreased trichome length are consistent with our previous studies
under different light levels without UVR (Cai et al., 2015). It has been reported that
enhanced UVR is one of the environmental factors that not only inhibit the growth of
cyanobacteria, but also change their morphology (Rastogi et al., 2014). Study showed
natural solar UVR would suppress formation of heterocysts and shorten the filament
length of Anabaena sp. PCC7120 (Gao et al., 2007). Natural levels of solar UVR in the
Sothern China were also found to break the filaments and alter the spiral structure of
*Arthrospira* (*Spirulina*) *platensis*, with a compressed helix that lessens UV exposures
for the cells (Wu et al., 2005). Cells in the trichomes of the estuarine cyanobacterium
*Lyngbya aestuarii* coil and then form small bundles in response to UV-B irradiation
(Rath and Adhikari, 2007). However, the shortened trichomes of *Trichodesmium* in this
work may be a result of UV-inhibited growth rather than a responsive strategy against
UV.



Carbon fixation in the long-term experiment showed similar patterns with the
short-term UV experiment, demonstrating that UV-A played a larger role in inhibiting
carbon fixation than UV-B. Since the ratio of UV-B to UV-A is lower in natural solar
light (1:50) than under our artificial UVR (1:28), the inhibitory effects of UV-B were
smaller compared to UV-A in the cultures under sunlight. Carbon fixation and $N_2$
fixation rates measured outdoors indicated that UV-induced carbon fixation inhibition
recovers quickly following transfer to PAR conditions, while the UV-induced $N_2$
fixation inhibition does not (Fig.6AC). Factors that might be responsible include lower
turnover rate of nitrogenase than that of RuBisco; more UV-induced damage to
nitrogenase with lower efficiency of repair (Kumar et al., 2003); and indirect harm
caused by ROS (Reactive Oxygen Species) induced by UV (Singh et al., 2014).
The UV effects in our study were measured under conditions that minimized self-
shading, namely during growth as single filaments. However, in its natural habitat
*Trichodesmium* often grows in a colonial form, with packages of many cells held
together by an extracellular sheath (Capone et al., 1998). In such colonial growth forms,
the effective cellular pathlengths for UV radiation are likely greatly increased, thereby
amplifying the overall sunscreen factor for the colony. *Trichodesmium*.spp might use
this colony strategy to protect themselves from natural UV damage in the ocean.
Our investigation shows that this cyanobacterium appears to have evolved the
ability to produce exceptionally high levels of UV protective compounds, likely
mycosporine-like amino acids. However, even this protective mechanism is insufficient
to prevent substantial inhibition of nitrogen and carbon fixation in the high-irradiance
environment where this genus lives. *Trichodesmium* spp are distributed in the upper
layers of the euphotic zone in oligotrophic waters, and its population densities are
generally greatest at relatively shallow depths (20 to 40 m) in the upper water column
(Capone et al., 1997). It seems likely that UV inhibition therefore significantly reduces
the amount of critical new nitrogen supplied by *Trichodesmium* to the N-limited
oligotrophic gyre ecosystems, a possibility that has not been generally considered in



regional or global models of the marine nitrogen cycle.
*Trichodesmium* can form dense, extensive blooms in the surface oceans, and a
frequently cited estimate of global nitrogen fixation rates by *Trichodesmium* blooms is
~42 Tg N $yr^{-1}$ (Westberry et al., 2006). Previous biogeochemical models of global $N_2$
fixation have emphasized controls by many environmental factors, including solar PAR
radiation, temperature, wind speed, and nutrient concentrations (Luo et al., 2014), but
have largely neglected the effects of UV radiation. When estimating $N_2$ fixation using
incubation experiments in the field, however, marine scientists have typically excluded
UV radiation by using incubation bottles made of UV-opaque materials like
polycarbonate (Olson et al., 2015). Our results suggest that under solar radiation at the
surface ocean, including realistic levels of UVR inhibition lowers estimates of carbon
fixation and $N_2$ fixation by around 47% and 65%, respectively (Fig.6).
Thus, it seems likely that shipboard measurements and possibly current model
projections of *Trichodesmium* $N_2$ fixation and primary production rates that do not take
into account UV inhibition could be substantial overestimates. However, our study was
only carried out under full solar radiation, simulating sea surface conditions, so further
studies are needed to investigate depth-integrated UV inhibition. Moreover, the
response to UV radiation may be taxon-specific. For example, unicellular $N_2$-fixing
cyanobacteria such as the genus *Crocosphaera*, with smaller cell size and thus greater
light permeability, may be more vulnerable to UV radiation than *Trichodesmium* (Wu
et al., 2015). In the future, as enhanced stratification and decreasing mixed layer depth
expose cells to relatively higher UV levels, differential sensitivities to UV radiation
may result in changes in diazotroph community composition. Such UV-mediated
assemblage shifts could have potentially major consequences for marine productivity,
and for the global biogeochemical cycles of nitrogen and carbon.

**Acknowledgements**





This study was supported by the National Key Research Programs 2016YFA0601400
and National Natural Science Foundation (41430967; 41120164007) to KSG, and by
U.S. National Science Foundation grants OCE 1260490 and OCE 1538525 to F-X.F.
and D.A.H. DAH and F-X.F.'s visit to Xiamen was supported by MEL's visiting
scientists programs. The authors would like to thank Nana Liu and Xiangqi Yi from
Xiamen University for their kind assistance during the experiments.


























**Figures**

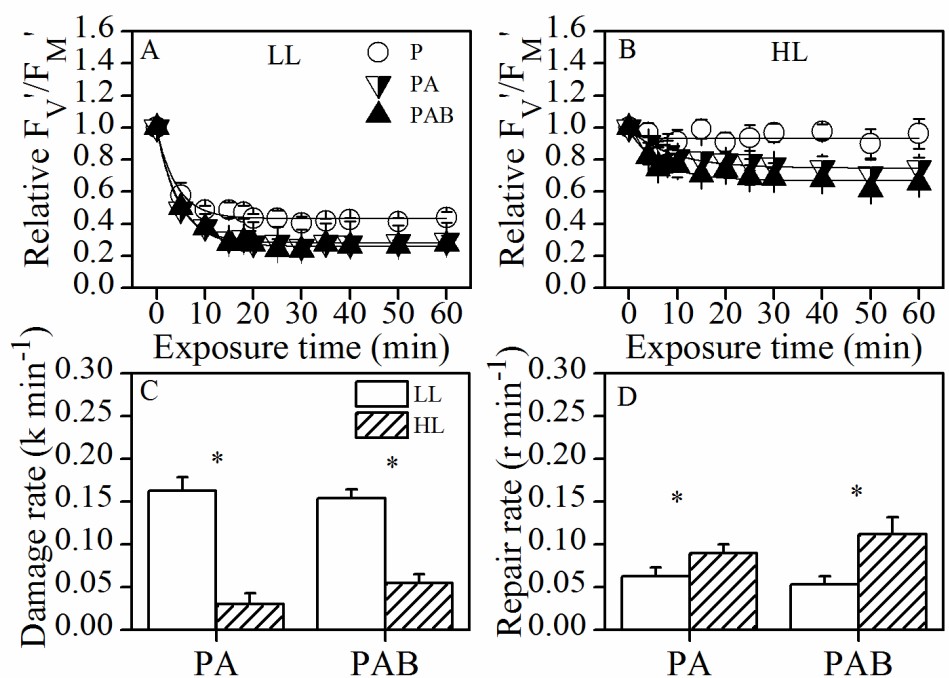

Fig.1 Changes of effective quantum yield ($F_V'/F_M'$) of *Trichodesmium* IMS101 grown

under (A) LL and (B) HL conditions while exposed to PAR (P), PAR+UVA (PA) and

PAR+UVA+UVB (PAB) under solar stimulator for 60 min. PSII damage (C; k, in $min^{-1}$) and repair rates (D; r, in $min^{-1}$) of LL- and HL-grown cells were derived from the

yield decline curve in the upper panels. Asterisks above the histogram bars indicate

significant differences between LL- and HL-grown cells. Values are the mean ±SD,

triplicate incubations.




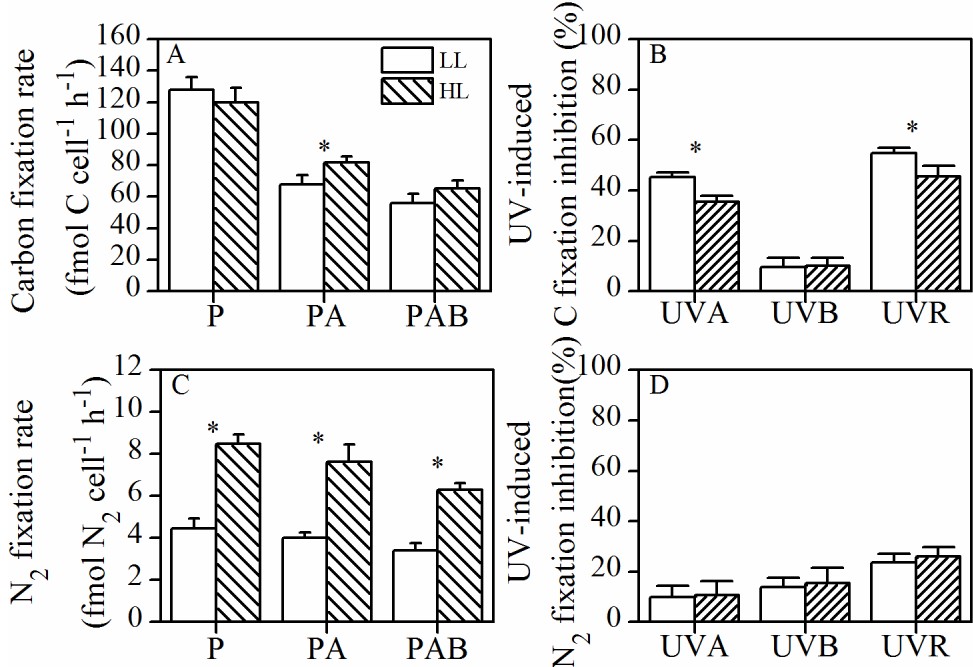


Fig.2 Photosynthetic carbon fixation rate (A; fmol C cell$^{-1}$ h$^{-1}$) and UV-induced C
fixation inhibition (B), N$_2$ fixation rate (C; fmol N$_2$ cell$^{-1}$ h$^{-1}$) and corresponding UV-
induced N$_2$ fixation inhibition (D) of *Trichodesmium* IMS101 grown under LL and HL
conditions. Asterisks above the histogram bars indicate significant differences between
LL- and HL-grown cells. Values are the mean ±SD, triplicate incubations.

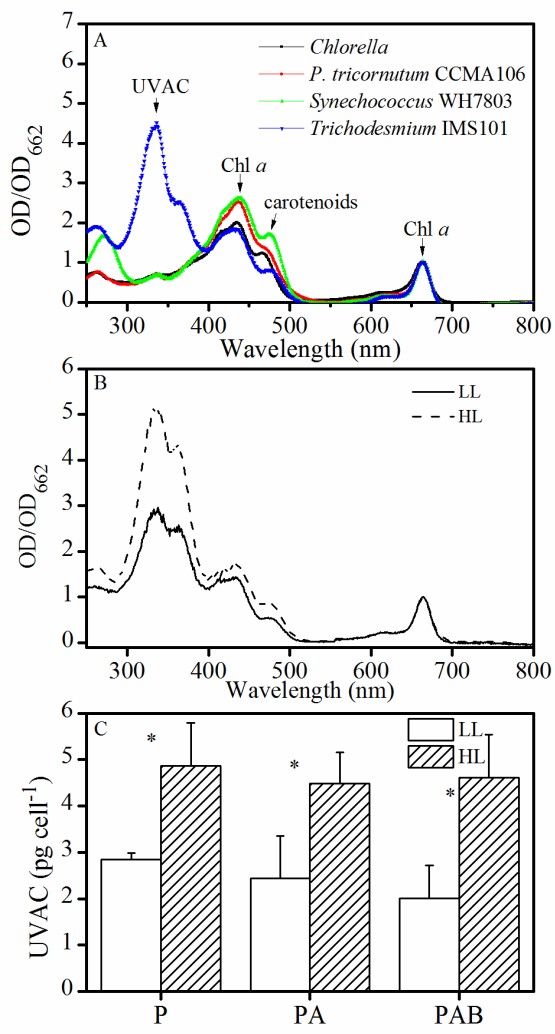


Fig.3 (A) Absorption spectrum of *Trichodesmium* IMS101 compared to other
phytoplankton. Pigments were extract by 100% methanol. OD value normalized to
$OD_{662}$ (Chl *a*). (B) Absorption spectrum of the *Trichodesmium* IMS101 grown under
LL and HL conditions, OD value normalized to $OD_{662}$ (Chl *a*). (C) Cellular contents of
UVACs of *Trichodesmium* IMS101 grown under LL and HL conditions after exposure
to PAR (P), PAR+UVA (PA), PAR+UVA+UVB (PAB) under solar stimulator for 10 h.
Asterisks above the histogram bars indicate significant differences between LL- and
HL-grown cells. Values are the mean ±SD, triplicate incubations.



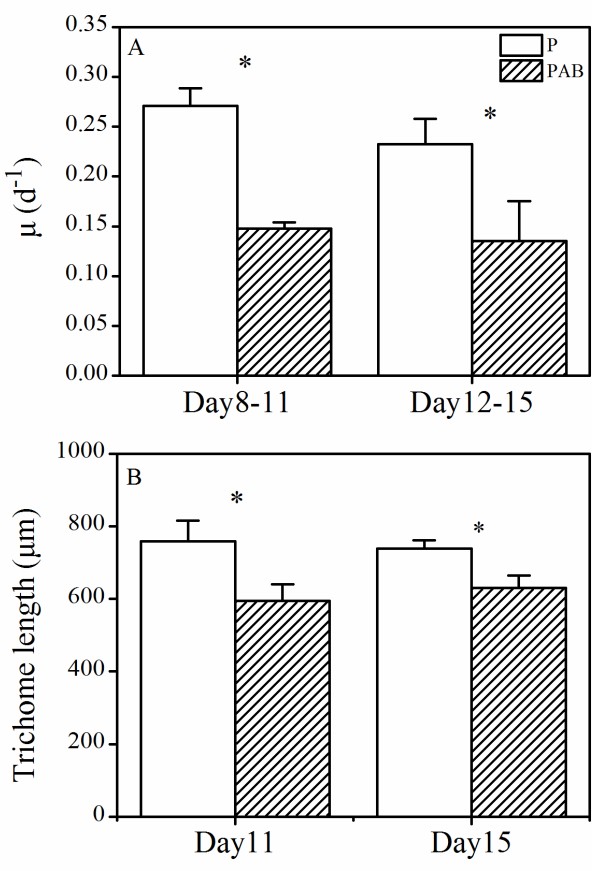

Fig.4 (A) Specific growth rate (measured during 8th-11th and 12th-15th day) of *Trichodesmium* IMS101 grown under solar PAR (P) and PAR+UVA+UVB (PAB). Corresponding total solar doses from Day 8 to Day 11 and from Day 12 to Day 15 were 17.03 and 18.51 MJ, respectively. (B) Trichome length (measured on the 11th and 15th day) of *Trichodesmium* IMS101 grown under solar PAR (P) and PAR+UVA+UVB (PAB). The asterisks indicate significant differences between radiation treatments. Values are the mean ±SD, triplicate cultures.





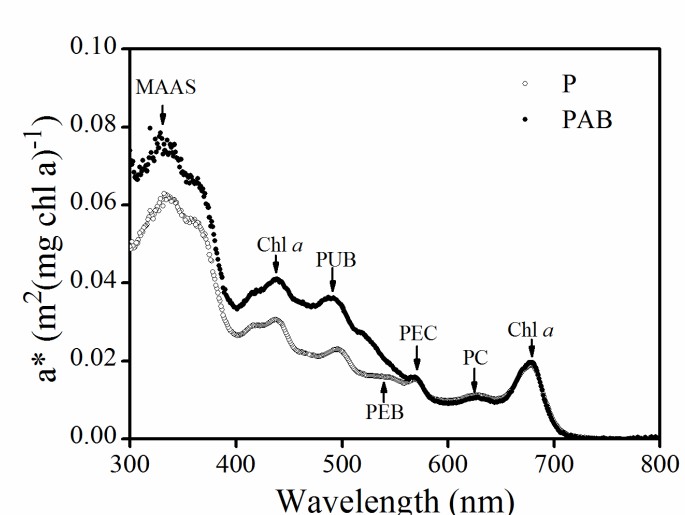


Fig.5 Chl *a* specific absorption spectrum (a$^*$) of *Trichodesmium* IMS101 grown under
solar PAR (P) and PAR+UVA+UVB (PAB). The measurements were taken on the 18th
day. The absorption peaks of MAAs (330 nm), PUB (495 nm), PEB (545 nm), PEC
(569 nm), PC (625nm) and Chl *a* (438 and 664 nm) are indicated.








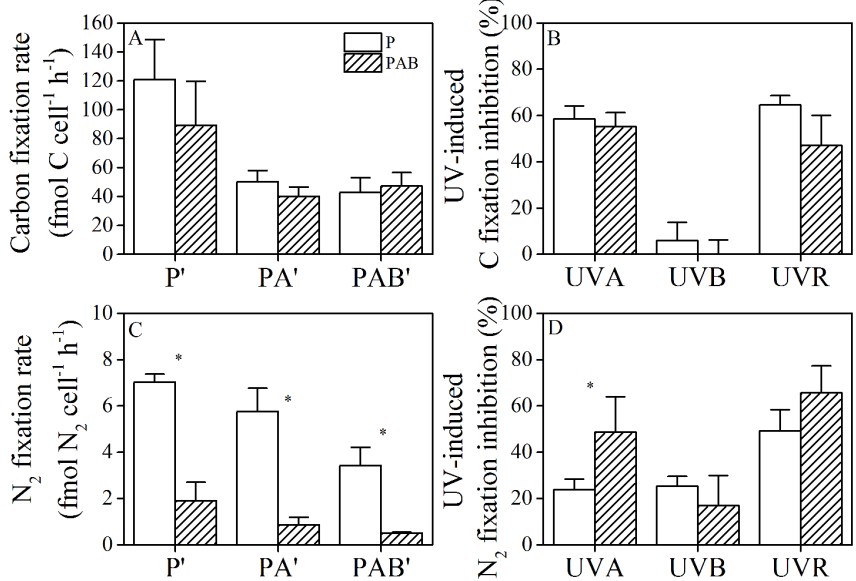

Fig. 6 Photosynthetic carbon fixation rate (A; fmol C cell$^{-1}$ h$^{-1}$) and UV-induced C fixation inhibition (B), N$_2$ fixation rate (C; fmol N$_2$ cell$^{-1}$ h$^{-1}$) and corresponding UV-induced N$_2$ fixation inhibition (D) of *Trichodesmium* IMS101 grown under solar PAR (P) and PAR+UVA+UVB (PAB). The measurement was taken on the 18[th] day at 11:00~13:00. Asterisks above the histogram bars indicate significant differences between P and PAB treatments. Values are the mean ±SD, triplicate incubations.



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
