# Peer review of "Discussion started: 24 March 2017 © Author(s) 2017. CC-BY 3.0 License."

_Biogeosciences, 2017_

## Referee Comment (RC1) · A. Banaszak (Referee) · 23 May 2017

General comments

This study explores the effects of ultraviolet radiation on a variety of physiological parameters in a marine, filamentous, nitrogen-fixing species of Trichodesmium that is important biogeochemically. Parameters measured include photochemistry, photosynthesis (as measured by carbon fixation), nitrogen fixation as well as chlorophyll concentrations and relative changes in ultraviolet absorbing compounds.

The studies were carried under both natural and artificial conditions using PAR (as a control) with PAR+UVA and PAR+UVA*UVB as treatments. Some experiments were

undertaken using short exposures and others using longer exposures that show that growth, photochemistry, photosynthesis and N2 fixation were negatively affected.

The importance of this work is that it shows that ultraviolet radiation affects a bloom-forming species and that prior ship-board measurements may have overestimated the nitrogen-fixing ability of this species by not taking into account UVR in their experimental design. Given that one of the effects of Climate Change and increase sea surface temperatures is a shallowing of the thermocline and thus cyanbacteria will have shorter recovery times form damage acuumulated while at the surface. Please see Helbling et al 2015 Scientific Reports). That study is relevant to the this one as the potential consequence is that Cyanobacteria may not be as successful in bloom production.

Overall, this work is well done and has a clear message. My only concern is the terminology used to describe the treatments: there is P, PA and PAB, P′, PA′ and PAB′and there is UVA, UVB and UVR. It is all rather confusing and I think maybe a small table in which you put the different acronyms with their respective meanings would be very useful. That or else in the methods section, clearly outline all of the different acronyms.

specific comments

I would move or remove the first paragraph about ozone depletion as it does not flow onto the rest of the introduction. The importance of CB is enough to justify the study.

Line 160-163: This movement of P, PA or PAB to "another treatment" – but which? You do not specify and this is very confusing. Then in Figure 2 on Carbon and N2 fixation you also have UVA, UVB and UVR and I have no idea what they represent in terms of your treatments. Lines 307 to 324 that detail the results using P, PA, PAB, P′, PA′, PAB′ and UVA, UVB and UVR are all confusing.

Figure 1: why are there no damage and repair rates for P treatment? Values for all three treatments are given in the text (lines 259 to 262) but not in Figure 1C.

Figure 2: For both carbon fixation and N2 fixation you calculated inhibition induced by UVA, UVB and UVR and termed this IP, IPA and IPAB. Why not use this terminology in the Figures 2B, 2D? – instead you use two different namings –UVA, UVB and UVR– this is confusing.

Line 274: In your UVB treatment, it includes UVA, right so the treatment is actually UVA+UVB?

Line 282: "other phytoplankton" is referred to here and in Figura 3A. Given that you are comparing with other cultures, you need to specify in the methods how they were grown and give their full names as they are abbreviated in the Figure itself. Some readers may not be aware of these species.

Line 294: "addition of UVR significantly reduced the trichome length by 22% and 11%" How can one treatment (UVR) cause two different reductions (22 and 11%)?

Lines 366-368: I think you should cite Neale et al 1998 J Phycol here.

One aspect that should be discussed more is that fact that UV absorbing compounds (most likely MAAs) are expensive to make (see Litchman et al 2002) in terms of Nitrogen in particular so this is an interesting aspect that should be discussed given your results. At the end of the paragraph (lines 465-467) would be a good place.

Technical corrections

Line 31: I suggest changing "especially" to "specifically" Line 34: change "lived" to "live" and "mixing" to "mixed" Line 35: change "UV radiation" to "UV-B radiation" and delete "especially UV-" Line 36: delete "B" and change "its" to "their" Line 98: change "was" to "and" Line 138, 139 Should "W" be "Wm-2"? Line 148: "Cell" should be "Cells" Line 150: delete "respective" Line 169: "interval" should read "inervals" Line 187, 200: "GF/filters" should read "GF/F filters"? Line 228, 229, 230: "inducted" should read "induced" Line 237: "radiation conditions" should read "radiation condition" Line 255: add "the" between "than PAR" Line 264: "treatment" should read "treatments"

Line 410: "consisted with" should read "was consistent with" Line 429: delete "Study showed" Line 430: Change "natural solar UVR would" to "natural solar UVR can" Line 431: Anabaena in italics Line 432: Sothern should read Southern

---

## Author Comment (AC1) · 10 Jun 2017

1.Comments: I would move or remove the first paragraph about ozone depletion as it does not flowbonto the rest of the introduction. The importance of CB is enough to justify the study.

Response:We have removed ozone depletion part.

2.Comments: Line 160-163: This movement of P, PA or PAB to "another treatment" – but which? You do not specify and this is very confusing. Then in Figure 2 on Carbon and N2 fixation you also have UVA, UVB and UVR and I have no idea what they represent in terms of your treatments. Lines 307 to 324 that detail the results using P', PA', PAB', P, PA, PAB and UVA, UVB and UVR are all confusing.

Response: We have added description in Line159-161:(namely P grown cells divided into P′, PA′, PAB′ treatments; PAB grown cells divided into P′, PA′, PAB′ treatments)

3.Comments: Figure 1: why are there no damage and repair rates for P treatment? Values for all three treatments are given in the text (lines 259 to 262) but not in Figure 1C.

Response: We added damage and repair rates of P treatment in Fig 1C and D.

4.Comments: Figure 2: For both carbon fixation and N2 fixation you calculated inhibition induced by UVA, UVB and UVR and termed this IP, IPA and IPAB. Why not use this terminology in the Figures 2B, 2D? – instead you use two different namings –UVA, UVB and UVR– this is confusing.

Response: We changed namings of UVA, UVB and UVR to IUVA, IUVB and IUVA+UVB in Fig 2 and Fig 6.

5.Comments: Line 274: In your UVB treatment, it includes UVA, right so the treatment is actually UVA+UVB?

Response: there actually three treatments in short-term exposure, one is PAR alone, one is PA (which is PAR+UVA), the other is PAB (which is PAR+UVA+UVB) treatment.

6.Comments: Line 282: "other phytoplankton" is referred to here and in Figura 3A. Given that you are comparing with other cultures, you need to specify in the methods how they were grown and give their full names as they are abbreviated in the Figure itself. Some readers may not be aware of these species.

Response: Yes, I have already given detail information about the full names of those species and growth conditions, which were written in Line 194-198 in M&M.

7.Comments: Line 294: "addition of UVR significantly reduced the trichome length by 22% and 11%" How can one treatment (UVR) cause two different reductions (22 and 11%)?

Respones: this experiment was conducted outdoor, light irradiance was different every day, so as the growth rate, the trichome length was measured on the day 11th and day 15th, on those two days, the trichome length of PAR+UVA+UVB treatment was reduced by 22% on day 11th and by 11% on the day 15th, compare to the PAR treatment. I have modified the sentence to make this statement more clearly in Line 294-295.

8.Comments: Lines 366-368: I think you should cite Neale et al 1998 J Phycol here. One aspect that should be discussed more is that fact that UV absorbing compounds (most likely MAAs) are expensive to make (see Litchman et al 2002) in terms of Nitrogen in particular so this is an interesting aspect that should be discussed given your results. At the end of the paragraph (lines 465-467) would be a good place.

Response: Thanks for your advice, I have cite this reference there. We added the citation in new Line 370-374: A red-tide dinoflagellate Gymnodinium sanguineum Hirasaka accumulates about 14-fold more MAAs in high (76 W•m-2) than in low (15 W•m-2) growth light and the high-light grown ones have lower sensitivity to UV radiation at wavelengths strongly absorbed by the MAAs (Neale et al., 1998). We also added new lines to discuss N limitation and UV sensitivity, in new Line 470-474: On the other hand, the UV absorbing compounds (most likely MAAs) are expensive to make in terms of nitrogen in particular (Singh et al., 2008). Decreased nitrogen supplied may increase sensitivity of phytoplankton assemblages to UV further (Litchman et al 2002), thus potentially creating a positive feedback between N-limitation and the UV sensitivity.

9.Comments: Technical corrections:

Response: All Revised

Please also note the supplement to this comment:
http://www.biogeosciences-discuss.net/bg-2017-106/bg-2017-106-AC1-supplement.pdf

[Figure]

**Supplement:**

[revised manuscript text omitted]

---

## Referee Comment (RC2) · S. Halac (Referee) · 3 Jul 2017

General Comments

The manuscript "Effects of ultraviolet radiation on photosynthetic performance and N2 fixation in Trichodesmium erythraeum IMS 101" describes very interesting work on the effects of UV radiation on bloom-formation cyanobacteria that contribuites to the input of N2 into oligotrophic sea waters (specially tropical and subtropical regions). The results on CO2 and N2 fixation decrease in cells exposed to UVR shows the importance of this study in a context of climate change as a larger proportion of the studied negative effects would increase under enhanced UVR doses. This increment would

be due to different factors, i.e., a more frequent stratification of the surface layer and a thermocline shallowing, both as a consequence of wáter temperature increase, and the higher UVR incidence on Earth surface because the ozone whole. Moroever, the study is well done and presented and only some changes in the manuscript need to be done. Therefore, publication of these data in Biogeosciences is fully justified. However, the addition of some important and clarifying paragraphs in some sections (mentioned below) is needed. Also the ecological consequences in a climate change context must be highlighted in the discussion section as well as including future research that would be necessary to confirm and/or deepen the consequences of the studied effects in C and N cycle on the ocean (see Trichodesmium ecological role as C and N source in the ocean, Berger et al., 2012).

Specific Comments

Introduction

The general objective of your investigation is not sufficiently justified, it would be better to connect your work with the need to investigate about the topics that are not explored yet (i.e., UVR effects on N2 fixation) and emphasize the importance of your results in the context of climate change. For example: Because of the importance of Trichodesmium in the input of carbon and nitrogen on oligotrophic oceans, and the lack of studies about the impact of enhanced UVR on the C and N fixation, is that we design experiments . . . . . . . . . . . . . . .. In particular, we evaluated the role of UVR in decreasing. . . . . . . . . The UVR doses we used represent realistic values in a current scenario (or future scenario of climatic change by the year . . . . . . . . .)

Material and Methods

1.-Line 87: I would replace "Estrategy Work" by "Experimental Design", and start explaining the experiments regarding the study's objective. For example, "The experiments to evaluate. . . . . . . . . . ..were carried on....... as follows:" 2.-Line 154: The specific growth rate is only calculated for days 8 to 11 and 12 to 16. What happened from days

1 to 7 is not shown, nor justified the reasons for that. If your study only assesed the exponential growth phase, it is necessary to define it. 3.-Line 167: The measurement of effective quantum photochemical yield is not justified. It would be clarifying to include a paragraph explaining what this proxy indicates. 4.-Line 199: Because the procedure for absorption spectra measurement is explained before for Trichodesmium, it's not necessary to repeat the same for the other species. 5.-Line 239: Acclimatization conditions of cultures instead of culture conditions is better understood

Results

1.-Line 286: Because UVACs values before the 10 hours exposure are not shown, it is not clear if the change is referred to time or to differences among PAB, PA and P. In this latter case, it would be better if you explained the idea in the following way: "did not present differences between radiation treatments after exposure………." 2.- Line 312: The paragraph is not clear and/or wrong because you talk about long-term UV-A exposure, and the long term treatments were only PAB and P, there was not PA. I would replace this paragraph with "inhibition induced by UV-A at short exposures in PAB and P acclimated cells. was......... and higher than inhibition induced by UV-B"

Discussion

1.- It would be necessary to give a better closure to the discussion adding future research (see General Comments) 2.- Lines 348, 431: The genus Anabaena for planktic morphotypes was replace by Dolichospermum since 2009 (see Wacklin et al., 2009) 3.-Line 412: I would replace "adaptation" with "acclimatization capacity depending on intensity and spectral quality of radiation". The latter is based on the difference between adaptation and acclimatization terms. 4.-Line 429: See Fiorda et al., 2011. It would be very valuable adding their results in the discussion about the change of morfology due to UVR exposure

Technical corrections

1.- Lines 255, 293, 303, 304: Be consistent in the used nomenclature, PAR treatment is already defined as P, and UVR treatment as PAB, so use the same terminology for all the cases 2.- Lines 266, 271, 287, etc: As was mentioned above: ultraviolet radiation is abreviated as UVR, use always the same 3.- Line 277: As was mentioned above: high light acclimated cells: HL; low light acclimated cells: LL 4.- Line 413: Change to Trichodesmium instead of Trichodemium 5.- Line 472: Remove radiation, PAR already includes this term

Please also note the supplement to this comment:
https://www.biogeosciences-discuss.net/bg-2017-106/bg-2017-106-RC2-supplement.pdf

---

## Author Comment (AC2) · 5 Jul 2017

General Comments: Also the ecological consequences in a climate change context must be highlighted in the discussion section as well as including future research that would be necessary to confirm and/or deepen the consequences of the studied effects in C and N cycle on the ocean (see Trichodesmium ecological role as C and N source in the ocean, Berger et al., 2012).

Response: we added discussion in Line 510-511: "future research that would be necessary to confirm and/or deepen the consequences of UV effects in carbon and nitrogen cycle in the ocean."

[Figure]

Specific Comments

Comments:Introduction The general objective of your investigation is not sufficiently justified, it would be better to connect your work with the need to investigate about the topics that are not explored yet (i.e., UVR effects on N2 fixation) and emphasize the importance of your results in the context of climate change. For example: Because of the importance of Trichodesmium in the input of carbon and nitrogen on oligotrophic oceans, and the lack of studies about the impact of enhanced UVR on the C and N fixation, is that we design experiments : : :: : :: : :: : :: : :.. In particular, we evaluated the role of UVR in decreasing: : :: : :: : : The UVR doses we used represent realistic values in a current scenario (or future scenario of climatic change by the year : : :: : :: : :)

Response: we added texts in new line78-80: "Because of the importance of Trichodesmium in the input of carbon and nitrogen on oligotrophic oceans, and the lack of studies about the impact of enhanced UVR on the C and N fixation, is that we design the experiments."

Comments: Material and Methods 1.-Line 87: I would replace "Estrategy Work" by "Experimental Design", and start explaining the experiments regarding the study0s objective. For example, "The experiments to evaluate: : :: : :: : :: : :.were carried on....... as follows:"

Responses: We replace "study strategy" by "experimental design". And added texts in line 88-90: "The experiments to evaluate how UVR affects photosynthesis and N2 fixation of Trichodesmium were carried on indoor and outdoor as follows:"

2.-Line 154: The specific growth rate is only calculated for days 8 to 11 and 12 to 16. What happened from days 1 to 7 is not shown, nor justified the reasons for that. If your study only assesed the exponential growth phase, it is necessary to define it.

Responses: We added texts to explain it in new line 154-155: "In order to evaluate adaptation responses of Trichodesmium to natural solar irradiance, all parameters were obtained after one week acclimation outdoor."

3.-Line 167: The measurement of effective quantum photochemical yield is not justified. It would be clarifying to include a paragraph explaining what this proxy indicates.

Responses: we added texts to explain Fv'/Fm' in line 173-175: "Effective photochemical quantum yield (FV'/FM') is generally considered to be light quantum using efficiency. We use this parameter to indicate Photosystem II activity."

4.-Line 199: Because the procedure for absorption spectra measurement is explained before for Trichodesmium, it's not necessary to repeat the same for the other species.

Responses: we added text "as the same method in Trichodesmium" in lin 208 to illustrate the same measurement as Trichodesmium. But in the Trichodesmium part I emphasize the Chlorophyll-specific absorption cross-sections (a*) measurements not the Chl a measurement.

5.-Line 239: Acclimatization conditions of cultures instead of culture conditions is better understood

Responses: revised in new line 247.

Comments:Results 1.-Line 286: Because UVACs values before the 10 hours exposure are not shown, it is not clear if the change is referred to time or to differences among PAB, PA and P. In this latter case, it would be better if you explained the idea in the following way: "did not present differences between radiation treatments after exposure: : :: : :: : :."

ResponseïijŽwe added texts : "….not present differences between radiation treatments after exposure to UV for 10 hrs." in line 295.

2.- Line 312: The paragraph is not clear and/or wrong because you talk about long-term UV-A exposure, and the long term treatments were only PAB and P, there was not

PA. I would replace this paragraph with "inhibition induced by UV-A at short exposures in PAB and P acclimated cells. was......... and higher than inhibition induced by UV-B"

Responses: we revised the text to "However, inhibition induced by UV-A at short exposures was about 58% in both P and PAB treatments and significantly higher than inhibition induced by UV-B radiation (Fig. 6B, p<0.01)."in line 321-325.

Comments: Discussion 1.- It would be necessary to give a better closure to the discussion adding future research (see General Comments)

Responses: we emphasize the future research in the last paragraph: "….future research that would be necessary to confirm and/or deepen the consequences of UV effects in carbon and nitrogen cycle in the ocean." 2.- Lines 348, 431: The genus Anabaena for planktic morphotypes was replace by Dolichospermum since 2009 (see Wacklin et al., 2009) We added the new name in brackets in line 359 3.-Line 412: I would replace "adaptation" with "acclimatization capacity depending on intensity and spectral quality of radiation". The latter is based on the difference between adaptation and acclimatization terms.

ResponsesïijŽreplaced

4.-Line 429: See Fiorda et al., 2011. It would be very valuable adding their results in the discussion about the change of morfology due to UVR exposure

Responses:We added texts to show their discussion : "….... because UVR may affect calcium signaling then the expression of the key genes responsible for cell differentiation"

Technical corrections

Responses: All revised.

Please also note the supplement to this comment:
https://www.biogeosciences-discuss.net/bg-2017-106/bg-2017-106-AC2- supplement.pdf

**Supplement:**

[revised manuscript text omitted]